# Ada-iD: Active Domain Adaption for Intrusion Dection

## ABSTRACT

Vision-based intrusion detection has many applications in life environments, *e.g.*, security, intelligent monitoring, and autonomous driving. Previous works improve the performance of intrusion detection under unknown environments by introducing unsupervised domain adaption (UDA) methods. However, these works do not fully fulfill the practical requirements due to the performance gap between UDA and fully supervised methods. To address the problem, we develop a new and vital **a**ctive **d**omain **a**daption intrusion detection task, namely ADA-ID. Our aim is to query and annotate the most informative samples of the target domain at the lowest possible cost, striving for a balance between achieving high performance and keeping low annotation expenses. Specifically, we propose a multi-task joint active domain adaption intrusion detection framework, namely ADAID-YOLO. It consists of a lower branch for detection and an upper branch for segmentation. Further, three effective strategies are designed to better achieve the ADA-ID task: 1) An efficient **D**ynamic **D**iffusion **P**seudo-**L**abeling method (DDPL) is introduced to get Pseudo ground truth to help identify areas of uncertainty in segmentation. 2) A **E**nhanced **R**egion **I**mpurity and **P**rediction **U**ncertainty sampling strategy (Enhanced-RIPU) is proposed to better capture the uncertainty of the segmentation region. 3) A **M**ulti-**E**lement **J**oint sampling strategy (MEJ) is designed to calculate the uncertainty of the detection comprehensively. Finally, comprehensive experiments and comparisons are conducted on multiple dominant intrusion detection datasets. The results show that our method can outperform other classic and promising active domain adaption methods and reach current SOTA performance, even surpassing the performance of UDA and full supervision on Normal→Foggy with only 0.1% and 10% data annotation, respectively. All the source codes, and trained models will be public.

## CCS CONCEPTS

• **Computing methodologies → Activity recognition and understanding**; Computer vision tasks.

## KEYWORDS

Active domain adaption intrusion detection, Framework, Active sampling strategy

## 1 INTRODUCTION

With the development of society, vision-based intrusion detection has many applications in daily life, such as in security, autonomous

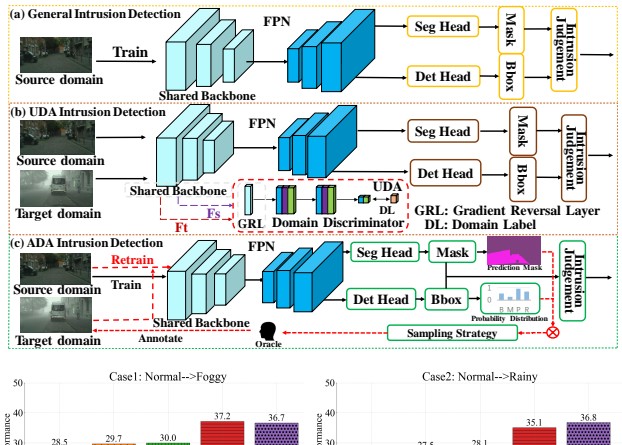

**Figure 1: The workflow and performance comparison of different intrusion detection paradigms. Here, we present three different paradigms, general intrusion detection [30, 33], unsupervised domain adaption (UDA) intrusion detection [10], and active domain adaption (ADA) intrusion detection, respectively. We can find that the UDA methods can help to improve intrusion detection performance, N→F: +1.2%, N→R: +7.5%. However, compared with fully supervised (Oracle), the performance gap also exists, N→F: -7.0%, N→R: -9.3%. Interestingly, our sampling approaches can effectively improve the performance of intrusion detection, even surpassing the performance of UDA and full supervision on Normal→Foggy with only 0.1% and 10% data annotation.**

driving, and intelligent video surveillance. Vision-based intrusion detection revolves around the evaluation of whether a potential object is present within a specific restricted area-of-interest (AoI). Based on whether the camera is moving, intrusion detection can be divided into static and dynamic intrusion detection. For static-view intrusion detection, some promising works and algorithms, *e.g.*, conditional random field (CRF) [21], Histogram of Oriented Gradients (HOG) [44] and background subtraction [24], are proposed. However, most of these works cannot meet the needs of real-time intrusion detection under dynamic view due to their simplicity. To tackle the issue of *real-time* and *accuracy* in intrusion detection under dynamic view, some encouraging works are designed and proposed. PIDNet [33] and Cross-PIDNet [30], are first proposed to solve the problem of pedestrian intrusion detection in dynamic view. These encouraging works greatly boost the performance of dynamic-view intrusion detection tasks to some extent. However, the performance of intrusion detection will degrade when generalizing these well-trained models in normal weather to an unknown environment or

domain *e.g.*, Foggy, Rainy, Night. A significant reason is that the domain shift remains between the normal weather (source domain) and unknown environment (target domain) [4, 11, 18, 46].

Benefiting from unsupervised domain adaption (UDA), some promising unsupervised domain adaption methods are proposed to solve the problem of domain shift between source and target domain, *e.g.*, DANN [9], CDAN [18], JAN [19]. Based on this method, MMID-bench [10] proposes a Multi-domain, Multi-category intrusion detection task to solve the problem of multi-category intrusion detection in challenging environments. These unsupervised adaption methods effectively avoid the problem of data labeling and improve the intrusion detection performance on unlabeled target domains. As shown in Figure 1. Compared with source only, the intrusion detection performance will boost when some promising UDA methods are used, *e.g.*, N→F: +1.2%, N→R: +7.5%. Nevertheless, compared with the Oracle (fully-supervised) result, the intrusion detection performance largely falls behind, *e.g.*, N→F: -7.0%, N→R: -9.3%.

To further reduce the intrusion performance between UDA and fully supervised, the natural idea is that, like ImageNet [6], collecting more data on different weather and labeling them. Then, we can use these labeled data to retrain the model and improve the model's generalization performance for intrusion detection in adverse weather. However, although this method is effective, the way of collection and labeling is *time-consuming* and very *expensive*. Based on these reasons, some promising active domain adaption (ADA) methods are proposed to further improve the generalization performance of the model under unknown environments [13, 20, 26, 32, 34, 38, 39, 42]. The purpose of ADA is to reduce the cost of labeling data and improve model generalization performance by selectively querying the most informative samples [17]. Currently, ADA achieves encouraging performance in multiple multimedia tasks, *e.g.*, image classification [36], semantic segmentation [23, 29, 37, 40], object detection [1, 5, 15, 25, 28, 41], image captioning [43], refractive error detection [8] and autonomous driving [12]. However, for intrusion detection tasks, active domain adaption still remains **blank** and **unexplored**. In this paper, we define this new and important task as dynamic-view **A**ctive **D**omain **A**daption **i**ntrusion **d**etection task, namely ADA-ID, for the first time.

To complete the dynamic-view ADA-ID task, a unified multi-task active domain adaption method, including an efficient and effective framework and corresponding sampling strategies, is currently lacking. Although some promising detection and segmentation networks are proposed [2, 3, 35, 45], these networks are not suitable for our ADA-ID task. The biggest reason is that our ADA-ID task is a joint task with detection and segmentation. To tackle the problem, we propose a unified, simple, yet efficient multi-task active domain adaption framework, namely ADAID-YOLO, with a lower branch for detection and an upper branch for segmentation. Besides, in order to better accomplish the ADA-ID task, three effective strategies are developed: 1) An efficient **D**ynamic **D**iffusion **P**seudo-**L**abeling method (DDPL) is introduced to get Pseudo ground truth to help identify areas of uncertainty in segmentation. 2) A **E**nhanced **R**egion **I**mpurity and **P**rediction **U**ncertainty sampling strategy (Enhanced-RIPU) is proposed to better capture the uncertainty of the segmentation region. 3) A **M**ulti-**E**lement **J**oint sampling strategy (MEJ) is designed to calculate the uncertainty of the detection

comprehensively. Finally, the total uncertainty is calculated by fusing the uncertainty of segmentation and detection. Experimental results denote that the proposed framework and three strategies are effective and can meet the requirements of the ADA-ID task.

In short, our main contributions are listed as follows:

- To the best of our knowledge, the task of **A**ctive **D**omain **A**daption **i**ntrusion **d**etection (ADA-ID) is developed for the first time. And a unified, simple, yet efficient multi-task active domain adaption end-to-end framework, ADAID-YOLO, is proposed to accomplish the task for the first time.
- Three effective approaches, **D**ynamic **D**iffusion **P**seudo-**L**abeling (DDPL), **E**nhanced **R**egion **I**mpurity and **P**rediction **U**ncertainty sampling strategy (Enhanced-RIPU), and **M**ulti-**E**lement **J**oint sampling strategy (MEJ) are proposed to calculate the joint uncertainty of segmentation and detection tasks of each sample for better achieving the ADA-ID task.
- The performance of various classic and state-of-the-art sampling strategies on the ADA-ID task is tested and reported. Comprehensive experiments and comparisons are conducted to demonstrate the effectiveness of the proposed framework and strategies. The results show that the proposed sampling strategies can not only reach the level of current SOTA but even surpass the performance of full supervision with only 10% data annotation.

## 2 RELATED WORK

● **Vision-Based Intrusion Detection.** Intrusion detection mainly focuses on two directions: static and dynamic view. For static intrusion detection, some promising works, *i.e.*, Histogram of Oriented Gradients (HOG) [44], adaptive background subtraction [31], conditional random field (CRF) [21], background subtraction [24], are proposed. However, most of these methods cannot meet the needs of real-time intrusion detection under dynamic view. The biggest reason is that intruding objects and cameras of dynamic view are constantly moving, which poses higher requirements for intrusion detection. Fortunately, with the rapid development of computer vision, some promising works and networks for dynamic-view intrusion detection are proposed, *e.g.*, PIDNet [33], and Cross-PIDNet [30]. These two encouraging works improve the performance of pedestrian intrusion detection under dynamic view greatly. However, the generalization, intrusion categories, and detection speed of these two works are insufficient. To tackle this issue, MMID-bench [10] first proposes a Multi-domain Multi-category intrusion detection task to further solve the problem of multi-category intrusion detection in challenging environments with the unsupervised domain adaptation method. Nevertheless, compared with fully supervised, previous works also remain shortcomings in two aspects, *inadequate generalization* and *low intrusion performance*. Therefore, in this paper, we develop a new active domain adaption intrusion detection task for the first time to improve the generalization and intrusion performance in unknown environments.

● **Active Domain Adaption.** Active Domain Adaptation (ADA) aims to help improve the model's generalization ability in the target domain by selecting the most informative target instances for annotation. Currently, some promising ADA methods are proposed and bring significant performance improvements in various fields, *e.g.*,

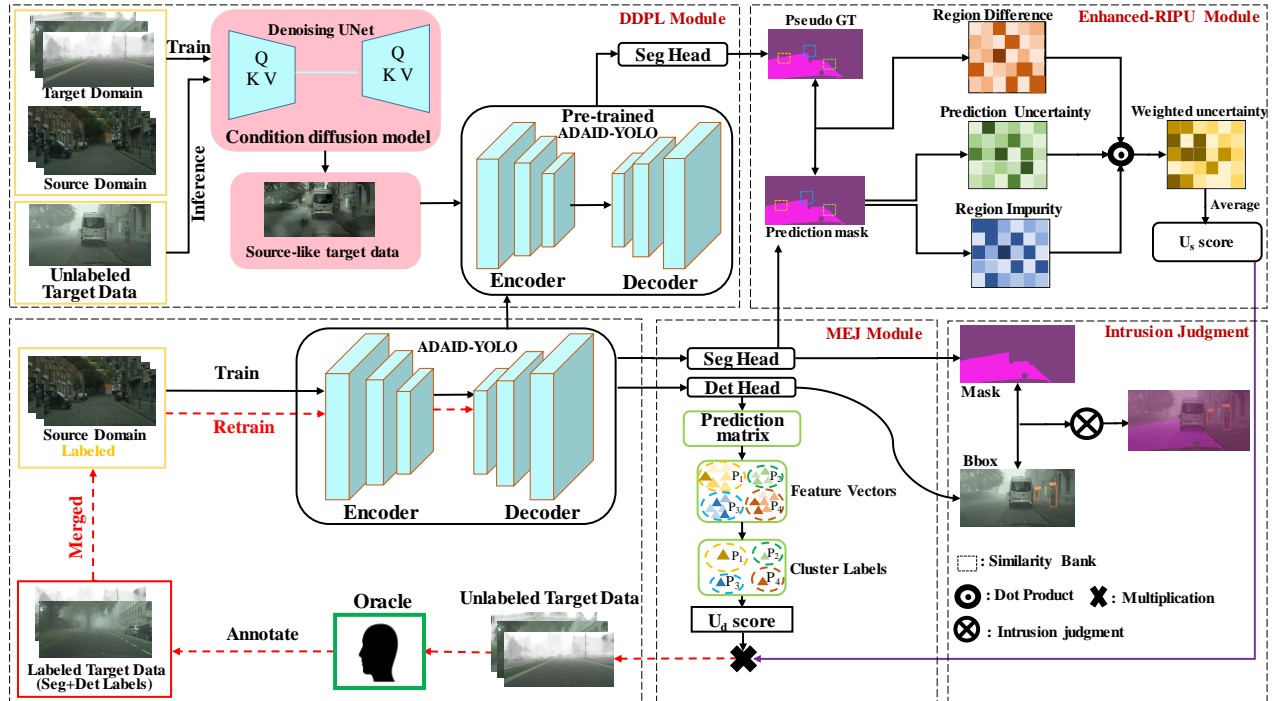

**Figure 2: Overall Architecture of the proposed ADAID-YOLO. We first use the original source domain (Normal-CMC) to train the model and validate its performance on the target domain (Target-CMC). Then, in every round, we utilize the proposed DDPL strategy to get the segmentation pseudo-GT ($\widetilde{GT}$) of target data. These $\widetilde{GT}$ are sent to the Enhanced-RIPU Module to compute the uncertainty of the segmentation branch. In addition, an MEJ Module is designed to calculate the uncertainty of the detection branch. Finally, the total uncertainty can be calculated by integrating the segmentation and detection uncertainty. Similar to the previous work [10], segmentation and detection results are used to jointly determine the final intrusion result. The categories and intrusion labels will be given.**

Energy-based [38], Transferable Query Selection [7], Distinctive Margin [39], RIPU [37], and Bi3D [42]. However, for the ADA tasks, these methods seem inappropriate. The main reason is that our ADA-ID task is a joint multi-task. Besides, although active domain adaption has made promising progress in various fields, for intrusion detection tasks, active domain adaption still remains blank and unexplored. In this paper, we develop a new and vital **a**ctive **d**omain **a**daption **i**ntrusion **d**etection task, namely ADA-ID, to improve intrusion detection performance in unknown environments.
• **Effective Design and Strategies.** To better accomplish our ADA-ID task, three effective sampling approaches are designed. 1) An efficient **D**ynamic **D**iffusion **P**seudo-**L**abeling strategy is introduced to get Pseudo ground truth to help identify areas of uncertainty in segmentation. Note that our method of generating pseudo ground truth is low cost and high performance. 2) To make up for RIPU work [37], a **E**nhanced **R**egion **I**mpurity and **P**rediction **U**ncertainty strategy is proposed to better capture the uncertainty of the segmentation region. 3) Different from previous works [15, 28, 41], a **M**ulti-**E**lement **J**oint sampling strategy is designed to calculate the uncertainty of the detection comprehensively. Multiple informative factors of affecting intrusion detection are considered , *e.g.*, inter similarity, categories imbalance, and tiny object.

## 3 APPROACH

### 3.1 Preliminary

**Problem Definition.** In **a**ctive **d**omain **a**daption **i**ntrusion **d**etection (ADA-ID) task, given a labeled source domain set (Normal weather), *i.e.*, $\mathbf{D}_s = \left\{ \left( x_i^s, y_i^s \right) \right\}_{i=1}^{n_s}$, and unlabeled target domain set (Adverse weather), *i.e.*, $\mathbf{D}_t = \left\{ x_j^t \right\}_{j=1}^{n_t}$. where $n_s$, $n_t$ denotes the number of labeled images and unlabeled images in the source/target domain, respectively. $y$ denotes the label of source domain images. Besides, we assume the annotation budget is $\mathbf{B}$ and $\mathbf{B} \ll n_t$. Like the active domain adaption paradigm, we set an initially empty labeled target dataset ($\bar{\mathbf{Z}}_t$). The $\bar{\mathbf{Z}}_t$ will be updated in the $\mathbf{R}$ round of the sampling process. In the $q$-th sampling round where $q \leq \mathbf{R}$, a subset $\triangle \mathbf{D}_t^q$ is selected from $\mathbf{D}_t/\bar{\mathbf{Z}}_t$ and labeled by an Oracle (human expert). Then, $\bar{\mathbf{Z}}_t$ will be updated as $\bar{\mathbf{Z}}_t \leftarrow \bar{\mathbf{Z}}_t \cup \triangle \mathbf{D}_t^q$. After $\mathbf{R}$ rounds of sampling, the number of data in $\bar{\mathbf{Z}}_t$ reaches the upper limit of annotation budget $\mathbf{B}$, *i.e.*, $\left|\bar{\mathbf{Z}}_t\right| = \mathbf{B}$. In our ADA-ID task, Our aim is to query and annotate the most informative samples ($\bar{\mathbf{Z}}$) from the unlabeled target pooling ($\mathbf{D}_t$) at the lowest possible cost, striving for a balance between achieving high performance and keeping low annotation expenses.

## 3.2 Motivation

Before diving into the details of the method we propose, we first explore a basic yet important question as motivates of our approach. *In the ADA-ID task, what are the key factors improving intrusion detection performance?* To get the answer, we rethink and analyze the reason from the intrusion definition. In the intrusion detection task, the final intrusion judgment is calculated by pixel points [10, 30, 33], the judgment way can be described as

$$\mathbf{J}_s = \begin{cases} \text{Intrusion} & \text{if } \mathbf{B} \cap \mathbf{S} > t \\ \text{No-Intrusion} & \text{if } \mathbf{B} \cap \mathbf{S} \le t, \end{cases} \quad (1)$$

where $\mathbf{J}_s$ denotes the final results of intrusion judgment. $\mathbf{B}$ denotes the prediction bounding box of the detection branch. Differently, to calculate the final intrusion results, we set all the pixels in the detection bounding box to $\mathbf{1}$. $\mathbf{S}$ denotes the segmentation results of the restricted area-of-interest (AoI). In our paper, the AoI denotes road. $t$ denotes the setting threshold. Inspired by previous works [10, 30, 33], we set the threshold as 20, *i.e.*, when the intersection of $\mathbf{B}$ and $\mathbf{S}$ is greater than the threshold, it is judged as an Intrusion ('Y'). Otherwise, it is judged as No-intrusion ('N'). From the intrusion detection definition view, we know that final intrusion detection performance is affected by two key factors, *the accuracy of the detection bounding box and AoI segmentation*, respectively. Thus, in the ADA-ID task, a valuable sample should be able to greatly improve both aspects of performance. Inspired by previous related work [25], [15], we define these two factors as *Detection Uncertainty* (**DU**) and *AoI Segmentation Uncertainty* (**SU**). In the ADA-ID task, for every image ($\mathbf{I}_t$) of unlabeled target pooling, we can define its total uncertainty as

$$\mathcal{M}^t(\mathbf{U}_d^t; \mathbf{U}_s^t \mid \mathbf{I}_t, \Theta^n) = \mathcal{M}(\mathbf{U}_d^t \mid \mathbf{I}_t, \Theta^n) \cdot \mathcal{M}(\mathbf{U}_s^t \mid \mathbf{I}_t, \Theta^n), \quad (2)$$

where $\Theta$ denotes the using multi-task framework, *i.e.*, proposed ADAID-YOLO, shwon in Figure 2. Note that the $n$ denotes the sampling round, *e.g.*, $1^{st}$, $2^{nd}$, $3^{rd}$, $\cdots$. The sampling round is determined by annotation budget $\mathbf{B}$. $\mathcal{M}^t(\mathbf{U}_d^t; \mathbf{U}_s^t)$ denotes the total uncertainty of image ($\mathbf{I}_t$) in our ADA-ID task. $\mathcal{M}(\mathbf{U}_d^t)$, $\mathcal{M}(\mathbf{U}_s^t)$ denotes the uncertainty of invader detection and AoI segmentation, respectively. Based on the above analysis, to get the uncertainty of detection and segmentation, three efficient approaches, including DDPL, Enhanced-RIPU sampling strategy, and MEJ sampling strategy, are proposed to calculate the joint uncertainty of each sample for better achieving the proposed ADA-ID task.

## 3.3 Dynamic Diffusion Pseudo-Labeling

To get Pseudo-Labelling ($\widetilde{GT}$) to help identify areas of uncertainty in segmentation, we propose a new efficient and low-cost pseudo-labeling generation strategy, namely Dynamic Diffusion Pseudo-Labeling (DDPL). The special process is shown in Figure 3. Our DDPL strategy can divided into three steps. Specifically, we first design and train a condition diffusion model using source/target domain ($\mathbf{D}_s \bigcup \mathbf{D}_t$). The aim is to transform the target domain into the source domain (Source-like target data) and then use the pre-trained ADAID-YOLO model for inference ($\mathbf{D}_t$) to obtain our pseudo ground truth. Note that our pre-trained ADAID-YOLO is trained in the source domain ($\mathbf{D}_t$). To evaluate the superiority of our proposed

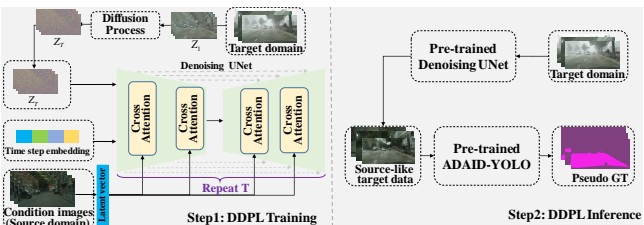

**Figure 3: The detailed pipeline of our DDPL strategy. Our DDPL strategy can be described as two main steps. Step 1: We train a well-designed condition diffusion model for learning the translation of the source domain ($\mathbf{D}_s$) and the target domain ($\mathbf{D}_t$). Step2: We use the pre-trained Denoising UNet model to infer input data from the target domain for obtaining the source-like target data, and use it to obtain the $\widetilde{GT}$.**

strategy, we compared our DDPL strategy with other methods, *i.e.*, larger UDA [10], promising zero-shot segmentation model: SAM [16]. The results of the ablation experiments show the effectiveness of our method, which not only generates correct pseudo ground truth but maintains a high inference speed. Detailed comparison experiment results can be seen in subsection 4.3.

## 3.4 Enhanced Region Impurity and Prediction Uncertainty sampling strategy

In our ADA-ID task, the segmentation branch aims to obtain accurate mask images of the restricted area-of-interest (AoI). However, the segmentation performance of the AoI is not ideal due to domain shifts (gaps) between the source domain ($\mathbf{D}_s$) and the target domain ($\mathbf{D}_t$). These domain shift can cause two phenomena: 1) when the invader encounters the AoI, it will be judged as non-intrusion ('N') due to inferior segmentation. But, the fact is that the intruder may caused a serious intrusion ('Y'). 2) Since the ADA-ID task is very sensitive to pixels, some erroneous judgments are exited in the edge region. These erroneous samples is defined as '*difficult samples*', which are important factors hindering the intrusion performance. Therefore, the sampling strategy is to select informative samples that can improve the performance of '*difficult samples*'. In order to query the informativeness, we propose a Enhanced Region Impurity and Prediction Uncertainty sampling strategy to better capture the uncertainty of the segmentation region.

Given an input image $\mathbf{I}_t$, and $\mathbf{I}_t \in \mathbb{R}^{\mathcal{H} \times \mathcal{W} \times C}$, where $\mathcal{H}$, $\mathcal{W}$ and $C$ denotes the width, height and channel dimension of input image $\mathbf{I}_t$, respectively. We define the segmentation branch model in our ADAID-YOLO as $\Theta_s$. Then, the prediction matrix can be expressed as $\mathbf{P}_t$. We can know that $\mathbf{P}_t = \Theta_s(\mathbf{I}_t)$, and $\mathbf{P}_t \in \mathbb{R}^{\mathcal{H} \times \mathcal{W} \times \mathcal{N}}$, where $\mathcal{H}$, $\mathcal{W}$, and $\mathcal{N}$ denote the width, height, and number of classes, respectively. To obtain our probability distribution, we conduct normalization for all pixels of $\mathbf{P}_t$, shown as $\tilde{\mathbf{P}}_t = \text{softmax}(\mathbf{P}_t)$. Besides, we can get the prediction pseudo label matrix by $\hat{\mathbf{P}}_t^{(i,j)} = argmax_{n \in \{1, \cdots N\}} \tilde{\mathbf{P}}_t^{(i,j,n)}$. where $n$ denotes the category. In our ADA-ID task, our segmentation task is to segment the restricted area-of-interest (AoI). Therefore, two classes exist for the segmentation branch, Class $\mathbf{0}$: background and Class $\mathbf{1}$: road. In

our work, to calculate our input image $\mathbf{I}_t$ uncertainty, we need to divide the input image $\mathbf{I}_t$ into multiple regions. As shown in Figure 4. The pixel region is defined as

$$\mathcal{M}_z(i,j) = \{(x,y)\,||x-i|\le z, |y-j|\le z\}, \qquad (3)$$

where $z$ denotes the size of the defined region and pixel region complies with $k$-$square$-$neighbors$, i.e., $(2z+1, 2z+1)$. Then, the region impurity can be expressed as

$$Q^{(i,j)} = -\sum_{n=1}^{N} \frac{|\mathcal{M}_z^n(i,j)|}{|\mathcal{M}_z(i,j)|} log \frac{|\mathcal{M}_z^n(i,j)|}{|\mathcal{M}_z(i,j)|}, \qquad (4)$$

where $|\mathcal{M}_z^n(i,j)|$ denotes the piexl number of category $n$ in the set. Besides, the region prediction uncertainty can be written as

$$\mathcal{U}^{(i,j)} = \frac{1}{|\mathcal{M}_z(i,j)|} \sum_{(x,y)\in\mathcal{M}_z(i,j)} \mathcal{S}^{(x,y)}, \qquad (5)$$

where $\mathcal{S}^{(i,j)} = -\sum_{n=1}^{N}\mathbf{P}_t^{(i,j,n)}log\mathbf{P}_t^{(i,j,n)}$. Based on the above analysis, to measure the uncertainty of input image $\mathbf{I}_t$, we proposed the Region difference to guide the calculation of region uncertainty and difference. Region difference denotes the difference between prediction and the pseudo ground truth. As shown in Figure 4. The calculation method is defined as

$$\mathcal{D}^{(i,j)} = \left\{(x,y)\in\mathcal{M}_z(i,j) \mid \left\langle \hat{\mathbf{P}}_t^{(x,y)} \ne \widetilde{\mathbf{GT}}^{(x,y)} \right\rangle \right\}, \qquad (6)$$

where $\widetilde{\mathbf{GT}}$ denotes the generated pseudo ground truth by DDPL strategy. $\langle\cdot\rangle$ denotes the calculation way of difference. Finally, the total uncertainty of input image $\mathbf{I}_t$ is written as

$$\mathcal{M}(\mathbf{U}_s^t \mid \mathbf{I}_t, \Theta^n) = \frac{1}{\mathcal{H}\times\mathcal{W}} \sum_{i=1}^{\mathcal{H}}\sum_{j=1}^{\mathcal{W}} Q^{(i,j)} \odot \mathcal{U}^{(i,j)} \odot \mathcal{D}^{(i,j)}, \quad (7)$$

where $\mathcal{H}$, $\mathcal{W}$ denotes the width and height of predicted mask images. $\odot$ denotes the dot product.

## 3.5 Multi-Element Joint sampling strategy

The detection branch provides accurate localization and category identification in intrusion detection tasks. Although previous works involved Class imbalance [15, 28, 41], the Inter-class similarity and Tiny objects still are not considered. To solve the issue, we propose the Multi-Element Joint sampling strategy to calculate $\mathcal{M}(\mathbf{U}_d)$. We first define the prediction matrix of detection branch as $\mathcal{X}$, where $\mathcal{X} \subset \mathbb{R}^{\mathcal{B}\times\mathcal{N}\times\mathcal{D}}$, $\mathcal{B}, \mathcal{N}, and \mathcal{D}$ denotes batch_size, the number of samples, the feature dimension [14]. And $\mathcal{N}$ can be express as

$$\mathcal{N} = \sum_{i=1}^{3} \frac{\mathbf{H}_s\cdot\mathbf{W}_s}{\mathbf{R}_i^2}\cdot\mathbf{N_f}, \qquad (8)$$

where $\mathbf{H}_s$ and $\mathbf{W}_s$ denote the reshaped height and width of input images. $\mathbf{R}_i$ denotes the three different downsampling rates, 8, 16, and 32, respectively. $\mathbf{N_f}$ denotes the number of anchors in every grid cell. Therefore, we can describe the final results matrix $\mathcal{X}$ as

$$\mathcal{X} = \begin{bmatrix} \mathbf{b}_{1,1} & \cdots & \mathbf{b}_{1,4} & \mathbf{c}_{1,5} & \mathbf{P}_{1,6} & \cdots & \mathbf{P}_{1,9} \\ \vdots & \vdots & \vdots & \vdots & \vdots & \vdots & \vdots \\ \mathbf{b}_{n,1} & \cdots & \mathbf{b}_{n,4} & \mathbf{c}_{n,5} & \mathbf{P}_{n,6} & \cdots & \mathbf{P}_{n,9} \end{bmatrix}, \qquad (9)$$

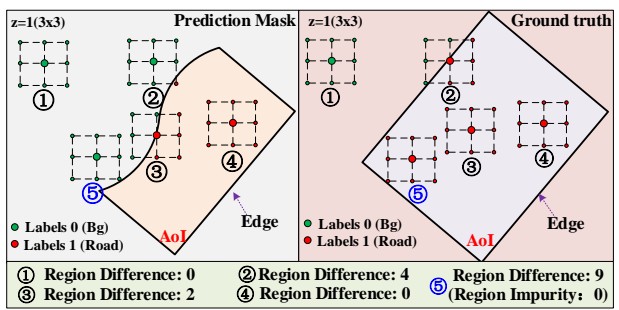

Figure 4: We show some cases between prediction mask and GT. Here, 'Bg' denotes background. ①, ②, ③, ④ denote four different cases, namely no intersection, partial intersection, the center point is on edge, and inside, respectively. We can find that the 'region difference' between the prediction mask and GT exists. This 'region difference' is one of the keys for measuring region uncertainty. Specifically, when the 'region difference' does not exist between them, we think the region is well-predicted and contains low uncertainty. The reverse is also true. In particular, for case ⑤, we find that, although the region impurity is zero [37], the region contains large uncertain information due to region difference (9). In fact, the region needs to be labeled and given a larger weight.

where the $\mathbf{b}_{i,j}$ represents the prediction bbox coordinates, the $\mathbf{c}_{i,j}$ represents the confidence score $\mathbf{c}$, and the $\mathbf{p}_{i,j}$ represent the probabilities distribution of the four different classes. In object detection, tiny objects usually present low confidence due to the difficulty of detection [35]. Besides, similar probabilities indicate high uncertainty. We extract confidence scores ($\mathbf{c}$) and probability distributions ($\mathbf{p}$) from $\mathcal{X}$ and express them as $\mathbf{F}_i = \left\{\mathbf{F}_c, \mathbf{F}_p\right\}$. Then the initial cluster assignments ($\mathcal{G}$) is calculated by spectral clustering [22] and is expressed as

$$\mathcal{G} = \mathcal{S}\left(\mathbf{N}_k \mid \mathbf{F}_c, \mathbf{F}_p\right), \qquad (10)$$

where $\mathcal{S}$ denotes the spectral clustering. $\mathbf{N}_k$ denotes the cluster number of confidence score and probabilities distribution. The detailed process can be found in **Appendix A**. To counteract the effects of class imbalance and enhance the subsequent uncertainty calculations, we employ a resampling method. The resampled representation, $\mathcal{G}'$, is generated by adjusting the sample distribution within each cluster, as expressed by $\mathcal{G}' = \mathbf{R}_s(\mathcal{G})$. $\mathbf{R}_s$ denotes the resampling. Building on the resampled dataset $\mathcal{G}'$, we compute the within-cluster confidence uncertainty, denoted as $C_k$, for each cluster $k$. This uncertainty is quantified by the standard deviation of the resampled confidence scores:

$$C_k = \sqrt{\frac{1}{\mathbf{N}_k - 1}\sum_{i=1}^{\mathbf{N}_k}\left(\mathbf{C}_k^i - \overline{\mathbf{C}_i}\right)^2}, \qquad (11)$$

where $C_k$ denotes the confidence uncertainty. $\overline{\mathbf{C}_i}$ and $\mathbf{C}_k^i$ denotes the confidence average and $i$-th resampled samples confidence scores in $k$-th cluster. Besides, the class probability distribution uncertainty can be calculated by

$$\mathcal{P}_k = -\frac{1}{|\mathbf{N}_k|} \sum_{i \in \mathbf{N}_k} \sum_{j=1}^{4} \mathbf{P}_{ij} \, log \, \mathbf{P}_{ij}, \tag{12}$$

where $\mathbf{N}_k$ is the number of re-samples in cluster $k$. $\mathbf{P}_{ij}$ denotes the the probability of the $j$-th category of the $i$-th samples. Therefore, the overall uncertainty score of input image $\mathbf{I}_t$ is the expressed as:

$$\mathcal{M}(\mathbf{U}_d^t \mid \mathbf{I}_t, \Theta^n) = \frac{1}{\mathbf{N}_k} \sum_{k=1}^{\mathbf{N}_k} (C_k^t + \mathcal{P}_k^t), \tag{13}$$

where $C_k^t$, and $\mathcal{P}_k^t$ denotes uncertainty of confidence, probability distribution and tiny object of image $\mathbf{I}_t$, respectively.

---

**Algorithm 1** Ada-iD Sampling Strategy

---

**Require:** Multi-task network ADAID-YOLO: $\Theta$. Segementation branch $\Theta_s$. Detection branch $\Theta_d$. The $t$-th unlabeled sample $\mathbf{I}_t$ from unlabeled pooling $\mathbf{D}_t$, Labeled source domain $\mathbf{D}_s$, and the sub-round of $q$-th, max annotation budget $\mathbf{B}$.

**Ensure:** The selected target set $\Delta D_t^q$

1: **for** cycle $\leftarrow$ 1 to $\mathbf{B}$ **do**
2:     $\Delta D_t^q \leftarrow \emptyset$
3:     Trian multi-task network $\Theta$ with source domain $\mathbf{D}_s$.
4:     **for** $t = 1$ to $\mathbf{D}_t$ **do**
5:        Infer the target domain data $\mathbf{D}_t$ using the Pre-train $\Theta$.
6:        Segmentation branch prediction results: $\mathbf{P}_t$.
7:        Detection barch prediction results: $\mathcal{X}$.
8:        Divide $\mathbf{P}_t$ into multiple region according to Eq. (3).
9:        Calculate $\mathcal{M}(\mathbf{U}_s^t)$ according to Eq. (4)-Eq. (7).
10:       Calculate $\mathcal{M}(\mathbf{U}_d^t)$ using Eq. (10)-Eq. (13).
11:       Calculate total uncertainty $\mathcal{M}(\mathbf{U}_d^t; \mathbf{U}_s^t)$ using Eq. (2).
12:     **end for**
13:     Sort the resulting uncertainties.
14:     Selection of labels for informative samples.
15: **end for**
16: **return** Selected target subset $\Delta D_t^q$

---

## 4 EXPERIMENTS AND ANALYSES

### 4.1 Experimental Settings

*4.1.1 Implementation Details.* We conduct all the experiments on a computer with 8 NVIDIA GeForce RTX 2080Ti GPUs. For fairness, the epoch and batch size of all experiments are set to 150 and 24, respectively. The initial learning rate, momentum, and optimizer weight decay are set to 0.01, 0.937, and $5e^{-4}$, respectively. Note that because the number of our original unlabeled target samples is 2502, thus, when using the '0.1%' annotation budgets, we can inquiry $2502 \times 0.1\% \approx 3$ ($1^{st}$) the target images. More implementation details can be found in the **Appendix B**.

*4.1.2 Datasets.* We evaluate/report the performance of our strategies on four dominant intrusion detection datasets [10]. In these intrusion datasets, multiple different intrusion categories are provided, *e.g.*, *Pedestrian* (**P**), *Bicycle* (**B**), *Motorcycle* (**M**), and *Rider* (**R**), and multiple common adverse weathers (domains), *e.g.*, *Normal* (**N**), *Rainy* (**R**), *Foggy* (**F**), and *Night* (**Ng**). More details and visualization results are presented in the **Appendix C**.

*4.1.3 Metrics.* To conduct a comprehensive performance evaluation and comparison experiments, five different metrics are used to evaluate the effectiveness of the proposed strategy, mIOU [3, 45], mAP [2, 35], AccY, AccN, and Acc [10], respectively. In addition, inspired by Bi3D [42], we also report the Closed Gap. The Closed Gap can be expressed as $\triangle CG = \frac{|\mathbf{P}_o - \mathbf{P}_s| - |\mathbf{P}_o - \mathbf{P}|}{|\mathbf{P}_o - \mathbf{P}_s|}$, where $\mathbf{P}_o$, $\mathbf{P}_s$, $\mathbf{P}$ denotes the performance of Oracle (Fully-supervised), Source only and different sampling strategies, respectively.

*4.1.4 Baseline/Comparison Models.* To verify the effectiveness of the proposed strategies, we conduct comparisons experiments with multiple typical methods, including classic strategies (*e.g.*, *Random Sampling* (**RS**), *Least Confident* (**LC**) [36], *Margin Sampling* (**MS**) [27], *Entropy Sampling* (**ES**) [36]), promising sampling strategies, *e.g.*, **RIPU** [37], and UDA methods, *i.e.*, **JAN** [19]. More descriptions of these methods can be found in the **Appendix D**.

### 4.2 Main Results

*4.2.1 Compared with classic and promising works.* We first compare our methods with promising active sampling strategies and report the detailed results. As shown in Table 1. From Table 1, we can find that: 1) for intrusion detection performance, compared with source only, our strategies can surpass *8.1%* (N→R), *1.5%* (N→F), and *3.6%* (N→Ng) in different cross-domain tasks. 2) Compared with the promising strategy (RIPU [37]), the intrusion performance is improved by *2.7%* (N→R). 3) Compared with some classic sampling strategies, our sampling approaches present even better performance. In addition, our sampling strategies also can improve the detection performance greatly in some cross-domain tasks, *e.g.*, *1.8%* (N→F), *1.9%* (N→Ng), which denotes the effectiveness of the proposed strategies. More comparison results in different datasets, *e.g.* BDD-intrusion, can be seen in **Appendix E**.

*4.2.2 Compared with UDA works.* We deeply review some cross-domain intrusion detection works, and find that previous works of solving intrusion detection under adverse weather mainly focus on UDA methods [10]. We compare our strategy with the classic UDA methods, *e.g.*, JAN [19]. From Tabel 1, we can see that the performance of our strategy can surpass the promising UDA method when using only *0.1%* (3/2502) samples of the manually labeled target data (N→F: 29.7%→30.0%, N→R: 27.5%→28.1%), which demonstrates the superiority of our sampling approach. More comparative results with UDA methods are shown in **Appendix E**.

*4.2.3 Visualisation Comparisons.* Further, we present some visualization results to validate the effectiveness of our methods. As shwon in Figure 5. From Figure 5, we can see that our sampling strategies can help our model to recognize the most typical intrusion behaviors, and the performance of our strategies is better than other methods. More visualization results are shown in the **Appendix E**.

### 4.3 Ablation Experiment

We conduct sufficient experiments and provide analyses to further explore the effectiveness of proposed strategies.

*4.3.1 Dynamic Diffusion Pseudo-Labeling.* We first compare our DDPL method with some common and effective methods, *e.g.*, Larger UDA model [10], SAM [16], to evaluate the effectiveness of

| Task (Source→Target) | AL Strategy | Framework: ADAID-YOLO | | | | | | |
|---|---|---|---|---|---|---|---|---|
| | | Annotation budget: 0.1% (3/2502), $1^{st}$ | | | | | | |
| | | mIOU(%) | mAP@.5(%) | mAP@.5:.95(%) | AccY(%) | AccN(%) | Acc(%) | Closed Gap ΔmIOU(%)↑ / ΔmAP@.5(%)↑ / ΔAcc(%)↑ |
| N→R | Source Only | 78.3 | 23.8 | 10.4 | 21.3 | 19.7 | 20.0 | – |
| | Random | 84.1 | 26.0 | 10.7 | 28.6 | 21.0 | 22.6 | +32.4% / +11.6% / +15.5% |
| | Entropy [36] | 88.7 | 27.1 | 11.0 | 37.1 | 20.0 | 23.7 | +58.1% / +17.5% / +22.0% |
| | Margin Sampling [27] | 87.5 | 26.2 | 11.1 | 33.5 | 21.1 | 23.7 | +51.4% / +12.7% / +22.0% |
| | Least Confident [36] | 87.9 | 26.0 | 11.0 | 27.6 | 23.7 | 24.5 | +53.6% / +11.6% / +26.8% |
| | RIPU [37] | 88.6 | 28.3 | 12.1 | 38.4 | 21.9 | 25.4 | +57.5% / +23.8% / +32.1% |
| | JAN (UDA) [19] | 89.4 | 30.4 | 12.9 | 36.4 | 25.1 | 27.5 | +62.0% / +34.9% / +44.6% |
| | Ours | **90.6** | **28.9** | 12.6 | 36.8 | 25.7 | **28.1** | +68.7% / +27.0% / +48.2% |
| | Oracle (Fully-Supervised) | 96.2 | 42.7 | 18.3 | 40.9 | 35.7 | 36.8 | – |
| N→F | Source Only | 96.0 | 33.8 | 16.6 | 43.2 | 24.6 | 28.5 | – |
| | Entropy [36] | 96.1 | 34.6 | 16.6 | 41.6 | 25.5 | 28.9 | +14.3% / +9.2% / +4.9% |
| | Random | 95.8 | 33.9 | 16.2 | 42.5 | 25.5 | 29.1 | -28.6% / +11.5% / +7.3% |
| | RIPU [37] | 95.9 | 35.2 | 16.7 | 40.8 | 26.3 | 29.4 | -14.3% / +16.1% / +11.0% |
| | JAN (UDA) [19] | 96.1 | 33.8 | 16.6 | 42.6 | 26.2 | 29.7 | +14.3% / +0.0% / +14.6% |
| | Margin Sampling [27] | 96.2 | 34.5 | 16.8 | 42.4 | 26.5 | 29.9 | +28.6% / +8.0% / +17.1% |
| | Least Confident [36] | 95.9 | 35.4 | 17.0 | 42.8 | 26.4 | 29.9 | -14.3% / +18.4% / +17.1% |
| | Ours | **96.1** | **37.2** | 18.0 | 42.7 | 26.6 | **30.0** | +14.3% / +39.1% / +18.3% |
| | Oracle (Fully-Supervised) | 96.7 | 42.5 | 19.9 | 43.2 | 34.9 | 36.7 | – |
| N→Ng | Source Only | 94.0 | 36.3 | 15.7 | 42.3 | 26.7 | 30.0 | – |
| | Margin Sampling [27] | 94.3 | 37.6 | 16.3 | 47.3 | 25.3 | 30.0 | +11.5% / +16.0% / +0.0% |
| | Random | 94.8 | 37.3 | 16.6 | 46.5 | 26.7 | 30.9 | +30.8% / +12.3% / +11.8% |
| | Least Confident [36] | 94.2 | 37.1 | 16.4 | 40.6 | 28.3 | 30.9 | +7.7% / +9.9% / +11.8% |
| | Entropy [36] | 94.5 | 37.3 | 15.8 | 41.9 | 28.8 | 31.6 | +19.2% / +12.3% / +21.1% |
| | RIPU [37] | 94.8 | 37.8 | 16.9 | 47.9 | 28.9 | 32.9 | +30.8% / +18.5% / +38.2% |
| | Ours | **94.5** | **39.7** | 17.6 | 44.4 | 30.7 | **33.6** | +19.2% / +42.0% / +47.4% |
| | JAN (UDA) [19] | 95.0 | 37.7 | 17.1 | 47.2 | 32.3 | 35.5 | +38.5% / +17.3% / +72.4% |
| | Oracle (Fully-Supervised) | 96.6 | 44.4 | 20.2 | 45.1 | 35.6 | 37.6 | – |

**Table 1: The quantitative results on different adaptation scenarios under 0.1% ($1^{st}$) annotation budget. Here, 'N', 'R', 'F', and 'Ng' denote using different datasets: Normal-CMC, Rainy-CMC, Foggy-CMC, and Night-CMC, respectively. Source only denotes the training on the source domain and inference on the target domain. Oracle denotes training and inference on the target domain.**

the proposed DDPL strategies. (Our detailed comparison of three different methods is shown in the **Appendix F**). The results are shown in Table 2. From Table 2, we can find that our DDPL strategy can show the best performance (Pixel Accuracy: *97.68%*) and maintain the faster inference speed. Note that although SAM [16] presents an excellent ability of zero-shot for obtaining the pseudo-labeling, the effect of pseudo-labeling is not the best. The main reason is that extra prompts, *e.g.*, location of *points* or *boxes* are needed, and SAM has a slow inference speed, which is time-consuming. Besides, the edge segmentation effect is not very good. For our ADA-ID task, the edge segmentation of restricted AoI is quite essential.

*4.3.2* **Enhanced Region Impurity and Prediction Uncertainty sampling strategy.** We further test the effectiveness of Enhanced-RIPU. As shown in Table 3. Firstly, we compare our strategy with the baseline (S), original RIPU [37], and Enhanced-RIPU (Larger UDA) method. We can find that our Enhanced-GIPU strategy can achieve the best intrusion detection performance, *27.9%* Acc, and can surpass them by *7.9%*, *2.5%*, and *1.1%*, respectively. Besides, we

| Method | Model | Inference times (per images, s) | Information Leakage (Extra prompt) | Pixel Accuracy |
|---|---|---|---|---|
| SAM [16] | Transformer (Vit) | 0.231 | ✓ | 90.55% |
| Larger UDA Model [10] | CNN (w/DANN) | 0.009 | ✗ | 95.10% |
| DDPL (Ours) | Transformer+CNN | 0.006 | ✗ | 97.68% |

**Table 2: The comparison of different strategies of obtaining Pseudo-Labelling. Pixel accuracy denotes the proportion of correct pixels (predicted) of the total pixels.**

explore the impact of different pixel region **z** on our ADA-ID task comprehensively. We can see that the best intrusion performance can be reached when pixel region is set to $z = 2(5 \times 5)$. The main reason is that the larger **z** can not capture the feature details of mask images, while the smaller **z** fails to recognize the image edges.

*4.3.3* **Multi-Element Joint sampling strategy.** We further test the effectiveness of the proposed MEJ sampling strategy and report the performance. As are shown in Table 4. From Table 4, we can find

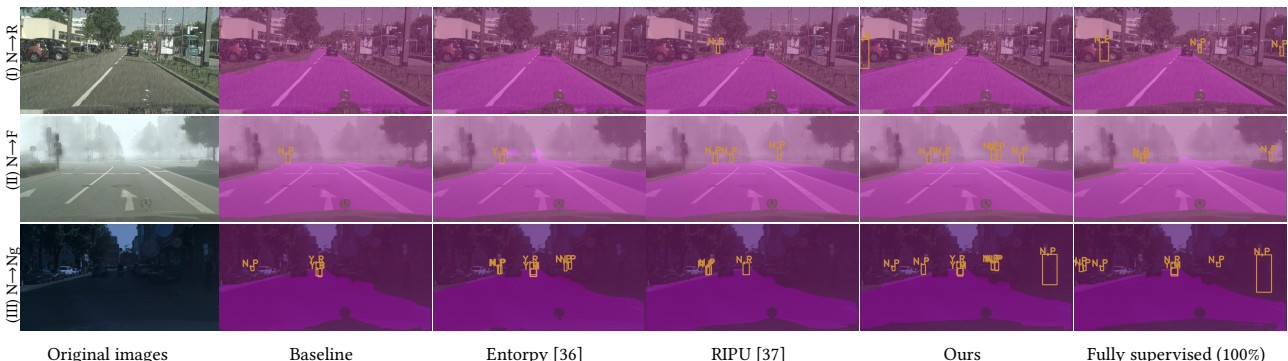

| | | | | | |
|---|---|---|---|---|---|
| Original images | Baseline | Entorpy [36] | RIPU [37] | Ours | Fully supervised (100%) |

**Figure 5: Some comparison of intrusion detection results. We can find that our strategy can help the model to detect the most typical intrusion behaviors and give correct labels. More visualization results are shown in the Appendix E.**

| Methods | | | | Metrics (Annotation budget: 0.1%, $1^{st}$) | | | | |
|---|---|---|---|---|---|---|---|---|
| Baseline | Enhanced-RIPU | FT | # z | mIOU(%) | mAP@.5(%) | AccY(%) | AccN(%) | Acc(%) |
| ✔(S) | ✘ | ✘ | - | 78.3 | 23.8 | 21.3 | 19.7 | 20.0 |
| ✔(R) | ✘ | ✘ | - | 84.1 | 26.0 | 28.6 | 21.0 | 22.6 |
| ✔[37] | ✘ | ⊙ | z=2(5×5) | 88.6 | 28.3 | 38.4 | 21.9 | 25.4 |
| ✔ | ✔† | ⊙ | z=2(5×5) | 89.8 | 28.7 | 33.1 | 25.1 | 26.8 |
| ✔ | ✔ours | ⊙ | z=2(5×5) | 89.3 | 29.4 | 31.3 | 27.0 | **27.9** |
| ✔ | ✔ours | ⊙ | z=3(7×7) | 89.1 | 28.0 | 27.3 | 26.0 | 26.3 |
| ✔ | ✔ours | ⊙ | z=1(3×3) | 89.4 | 28.2 | 32.3 | 25.2 | 26.7 |
| ✔ | ✔ours | ⊙ | z=2(5×5) | 89.3 | 29.4 | 31.3 | 27.0 | **27.9** |
| Oracle (Fully-Supervised) | | | | 96.4 | 41.9 | 39.9 | 36.3 | 37.0 |

**Table 3: The quantitative results of Enhanced-RIPU strategy. Task: N→R. 'S' and 'R' denote the results of the Source only and Random strategy. 'FT' denotes Fusion Type. † denotes the pseudo-labeling generated by the Larger UDA model.**

that, compared with baseline models (S) and (R), the MEJ strategy can surpass them by *6.4%* and *3.8%*, respectively. Besides, when different strategies are added, the performance of the intrusion will be improved, which proves the effectiveness of our MEJ strategy.

| Methods | | | | Metrics (Annotation budget: 0.1%, $1^{st}$) | | | | |
|---|---|---|---|---|---|---|---|---|
| Baseline | $C_k$ | $\mathcal{P}_k$ | $\mathcal{R}_s$ | mIOU(%) | mAP@.5(%) | AccY(%) | AccN(%) | Acc(%) |
| ✔(S) | | | | 78.3 | 23.8 | 21.3 | 19.7 | 20.0 |
| ✔(R) | | | | 84.1 | 26.0 | 28.6 | 21.0 | 22.6 |
| ✔ | ✔ | | | 84.7 | 26.5 | 35.8 | 20.1 | 23.4 |
| ✔ | ✔ | ✔ | | 86.8 | 26.2 | 28.2 | 24.0 | 24.9 |
| ✔ | ✔ | ✔ | ✔ | 87.9 | 28.6 | 33.2 | 24.6 | **26.4** |

**Table 4: The quantitative results of MEJ strategy. Task: N→R. $\mathcal{R}_s$ denotes the resample strategy. 'S' and 'R' denote the results of the Source only and Random strategy, respectively.**

## 4.4 More Insightful Experiment

*4.4.1 **Various target annotation budgets.*** We present some comparison results with various target annotation budgets to validate the superiority of our method. As shown in Figure 6. We can find that the performance of intrusion detection continues to rise

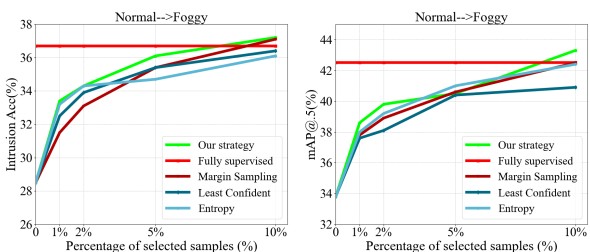

**Figure 6: The comparison results using different target annotation budgets. Task: N→F. We can find that our strategy outperforms the classical approach and can surpass fully supervised performance when annotation budgets reach 10%.**

as annotation budgets increase. Besides, our strategy consistently outperforms the classical approach for intrusion performance.

*4.4.2 **Compared with fully-supervised method.*** We also further explore the relationship between our method and fully supervised methods. The results are shown in Figure 6. From Figure 6, we can find that when the manually labeled target data reaches 10% of the total number of unlabeled samples, our strategy can significantly improve the cross-domain intrusion detection accuracy of the ADAID-YOLO detector, even surpassing the fully-supervised results with 100% labeled target data.

## 5 CONCLUSION

In this paper, we develop a new active domain adaption intrusion detection task, ADA-ID, for the first time. To accomplish this particular task, we first propose a multi-task active domain adaption framework, ADAID-YOLO. Besides, three effective methods, Dynamic Diffusion Pseudo-Labeling (DDPL), Enhanced Region Impurity and Prediction Uncertainty sampling strategy (Enhanced RIPU), and Multi-Element Joint sampling strategy (MEJ), are designed to better achieve the performance of the ADA-ID task. Comprehensive experimental results show that our proposed method can not only reach the level of the current SOTA but even surpass the performance of UDA and full supervision with only 0.1%, and 10% data annotation, respectively. In future work, we will further explore more efficient active domain adaptation strategies for ADA-ID tasks.

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
