# OpenReview forum: "Ada-iD: Active Domain Adaption for Intrusion Detection"
_acmmm.org/ACMMM/2024/Conference — MM2024 Oral_

### Official Review · Reviewer_utuT · 2024-05-24

**Rating:** 4
**Confidence:** 2

**Summary:**

The paper presents an unsupervised domain adaptation method for vision-based intrusion detection. Additional strategies are provided for pseudo-labeling and sampling, integrated in the framework and evaluated for several datasets.

**Strengths:**

The overall architecture is presented clearly, as well as the various improvements. The evaluations are thorough and sufficient comparisons to the state of the art are provided, with additional material in annex.

**Limitations:**

It is not clear to which use cases the architectural solution is tuned. In the paper, it is narrowed down to car/person detection in a static environment (a street), but the broader applicability (object detection? dynamic environments?) is unclear.

Minor comments:
- p.1: In the title, dection  -> detection; in the abstract: a enhanced -> an enhanced
- p.1: Both the abstract and intro start with the concept of "intrusions having many applications in real life". I would phrase this differently and more positively: the video-based applications are widespread, so care should be taken to avoid intrusions.
- p.1: The caption of some figures is huge and merely repeats what's in the text. E.g. for figure 1, the second half should be removed.
- p.2: "unsupervised domain adaption" is mentioned twice in "Benefiting from <...>".
- p.2: A large part of the abstract (on ADA-ID, ADAID-YOLO + 3 strategies) is literally repeated in the introduction (last paragraph) and in section 2 (first paragraph). The intro should be more about the general problem statement and background and less on related work.
- In general, avoid sentences starting with "And <...>" or "As shown <...>", but combine these with the previous sentence (throughout the whole paper).
- p.3: The variable x is not explained.
- p.3: ", Our" -> ", our"
- p.4: "as motivates of" -> "that motivates"
- p.4: Why is AoI always "road" in the paper? This restricts the use case a lot.
- p.4: "may caused" -> "may cause"; ". where" -> ", where" (several times throughout the paper)
- p.6: In Algorithm 1, correct "Trian", "Pre-train" and "barch". Rephrase "inquiry the target images". Shwon -> Shown.
- p.8: Entorpy -> Entropy
- Annex: check spelling mistakes (e.g. "segmentaion" several times)

**Suitability:**

2

---

### Official Review · Reviewer_12Ae · 2024-05-24

**Rating:** 5
**Confidence:** 3

**Summary:**

In this paper, the authors develop the Active Domain Adaptive Intrusion Detection (ADA-ID) task. And for the first time, a unified, simple, and efficient multi-task active-domain adaptive end-to-end framework, ADAID-YOLO, is proposed. In addition, three effective methods, Dynamic Diffusion Pseudo Labeling (DDPL), Enhanced Regional Impurity and Predictive Uncertainty Sampling Strategy (Enhanced RIPU), and Multi-Element Joint Sampling Strategy (MEJ), are designed to better achieve the performance of the ADA-ID task. The proposed framework is novel and experimentally detailed, the quality of the contributions and the presentation of the ideas as well as the analysis results is satisfactory, but there are still a series of problems that limit the application of the proposed framework.

**Strengths:**

This paper studied an interesting topic, and the mathematic works are sound, the three sampling methods can be considered efficient and reliable. The proposed method achieves a balance between high performance and keeping annotation costs low by querying and annotating the most informative samples in the target domain. In addition, this paper considers a variety of information factors affecting intrusion detection, such as similarity, category imbalance, tiny targets, etc., and verifies its adaptivity through simulation results.

**Limitations:**

1.	The authors should elaborate the derivations of several new results that are presented in the paper without reference or derivation (such as detection threshold and pseudo ground truth). If the derivations break the flow of the paper, they can be added to the annex.
2.	Although the contents related to motivation and the problem statement were convincing, the reviewer felt that the uncertainty sampling strategy process in the later section was a bit diluted.
3.	I do not think that the termination condition of the algorithm 3 is appropriate. Please check it again. Moreover, please check the format of formula (6) and (7). Moreover, the format of the references is inconsistent, some are abbreviations, but some are full names.
4.	The explanation and discussion of the results presented in the paper also needs improvements. The authors should have added a more detailed description of the experimental results to explain the trends, such as why the detection performance keeps improving in Figure 6.

**Suitability:**

2

---

### Official Review · Reviewer_ZxaS · 2024-05-25

**Rating:** 4
**Confidence:** 2

**Summary:**

This paper proposes ADA-ID, a novel approach to enhance intrusion detection systems by selectively querying and annotating the most informative samples from the target domain, thereby minimizing annotation costs and maximizing detection accuracy. It presents the ADAID-YOLO framework, which combines detection and segmentation tasks and employs three innovative strategies: Dynamic Diffusion Pseudo-Labeling (DDPL) to generate pseudo ground truth, Enhanced Region Impurity and Prediction Uncertainty (Enhanced-RIPU) to capture segmentation uncertainty, and Multi-Element Joint (MEJ) Sampling Strategy to calculate detection uncertainty comprehensively. Extensive experiments on various datasets demonstrate the effectiveness of ADAID-YOLO, achieving state-of-the-art performance with significantly reduced annotation costs.

**Strengths:**

1. The introduction of ADAID-YOLO, which combines detection and segmentation tasks, is a novel approach that enhances intrusion detection performance.
2. The proposed DDPL, Enhanced-RIPU, and MEJ strategies are innovative and effectively address the challenges of domain adaptation in intrusion detection.
3. The paper includes extensive experiments on multiple datasets, demonstrating the robustness and effectiveness of the proposed methods.
4. By reducing the annotation costs while maintaining high performance, the approach is practical and valuable for real-world applications.

**Limitations:**

1. It is important to explain N→F, N→R, and the introduction before the findings in the introduction, which is good for reviewers who are not from this background. Because Table 1 is on page 7.

2. In the abstract authors claimed that this application can be used for security, intelligent monitoring, and autonomous driving. In the experiment, I can only find the experiment for autonomous driving in the paper and appendix. Is there any other experiment that can prove the performance with the other two practical scenarios? It is not convincing enough to have multiple datasets to prove the real performance of the proposed method.

3. Ada-id in the title, and ADA-ID in the main content, authors should make the key name consistent.

**Suitability:**

3

---

### Meta-Review · Area_Chair_voLa · 2024-06-28

**Recommendation:** Accept (Oral)
**Confidence:** 5

**Metareview:**

The paper introduces ADA-ID, an innovative approach to enhance intrusion detection systems by selectively querying and annotating the most informative samples from the target domain. This minimizes annotation costs and maximizes detection accuracy. The proposed ADAID-YOLO framework combines detection and segmentation tasks and employs three novel strategies: Dynamic Diffusion Pseudo-Labeling (DDPL), Enhanced Region Impurity and Prediction Uncertainty (Enhanced-RIPU), and Multi-Element Joint (MEJ) Sampling Strategy. Extensive experiments on various datasets demonstrate the framework's effectiveness in achieving state-of-the-art performance with significantly reduced annotation costs.
The paper addresses a significant problem in the field of intrusion detection with a novel and practical approach. Despite some minor limitations and areas needing clarification, the strengths of the paper, including its innovative framework, effective strategies, and comprehensive experiments, make it a valuable contribution to the field.